# Automated Glaucoma Screening and Diagnosis Based on Retinal Fundus Images Using Deep Learning Approaches: A Comprehensive Review

**DOI:** 10.3390/diagnostics13132180

**Published:** 2023-06-26

**Authors:** Mohammad J. M. Zedan, Mohd Asyraf Zulkifley, Ahmad Asrul Ibrahim, Asraf Mohamed Moubark, Nor Azwan Mohamed Kamari, Siti Raihanah Abdani

**Affiliations:** 1Department of Electrical, Electronic and Systems Engineering, Faculty of Engineering and Built Environment, Universiti Kebangsaan Malaysia, Bangi 43600, Selangor, Malaysia; ahmadasrul@ukm.edu.my (A.A.I.); asrafmohamed@ukm.edu.my (A.M.M.); azwank@ukm.edu.my (N.A.M.K.); 2Computer and Information Engineering Department, College of Electronics Engineering, Ninevah University, Mosul 41002, Iraq; 3School of Computing Sciences, College of Computing, Informatics and Media, Universiti Teknologi MARA, Shah Alam 40450, Selangor, Malaysia; 2022236678@student.uitm.edu.my

**Keywords:** retinal fundus image, cup–disc ratio (CDR), glaucoma screening and diagnosis, deep learning, retinal disease, optic nerve head (ONH)

## Abstract

Glaucoma is a chronic eye disease that may lead to permanent vision loss if it is not diagnosed and treated at an early stage. The disease originates from an irregular behavior in the drainage flow of the eye that eventually leads to an increase in intraocular pressure, which in the severe stage of the disease deteriorates the optic nerve head and leads to vision loss. Medical follow-ups to observe the retinal area are needed periodically by ophthalmologists, who require an extensive degree of skill and experience to interpret the results appropriately. To improve on this issue, algorithms based on deep learning techniques have been designed to screen and diagnose glaucoma based on retinal fundus image input and to analyze images of the optic nerve and retinal structures. Therefore, the objective of this paper is to provide a systematic analysis of 52 state-of-the-art relevant studies on the screening and diagnosis of glaucoma, which include a particular dataset used in the development of the algorithms, performance metrics, and modalities employed in each article. Furthermore, this review analyzes and evaluates the used methods and compares their strengths and weaknesses in an organized manner. It also explored a wide range of diagnostic procedures, such as image pre-processing, localization, classification, and segmentation. In conclusion, automated glaucoma diagnosis has shown considerable promise when deep learning algorithms are applied. Such algorithms could increase the accuracy and efficiency of glaucoma diagnosis in a better and faster manner.

## 1. Introduction

Glaucoma is an eye disorder that is associated with the loss of retinal ganglion cells, whereby their axons gradually deteriorate over time, which leads to permanent vision loss if the condition goes untreated. Globally, 80 million individuals of different ages are affected by this disease, and it was considered the major cause of blindness in 2020 [1]. The main cause of this disease can be attributed to an imbalance between aqueous humor fluid drainage and flow that can result in increased intraocular pressure, which is a major risk factor for this disorder. The chance of getting glaucoma may additionally be increased by other elements such as age, race, and family history.

Extensive eye tests using tonometry, sight field tests, and an evaluation of the optic nerve head are crucial procedures for the diagnosis of glaucoma [2]. However, these tests usually take a lot of time, are costly, and require special equipment and expertise. Due to these limitations, there is an increasing trend in applying deep learning algorithms for automated glaucoma detection using fundus images.

Fundus imaging is a noninvasive modality that is easily accessible and provides essential information about the eye and optic nerve heads, including structural alterations used to indicate glaucoma presence. This image captures a detailed description of every aspect of the retina, including the size, shape, and color of significant regions such as the optic disc (OD), optic cup (OC), blood vessels, neuroretinal rim, and fovea [3,4]. Figure 1 shows the main structures in a retinal fundus image [5].

In the past few years, deep learning algorithms have shown good performance in diagnosing glaucoma based on fundus images, according to several studies that reported high sensitivity and specificity rates. However, this approach faces numerous challenges and constraints, such as the need to gather large and heterogeneous datasets, potential algorithm bias, and bureaucracy issues in healthcare system validation [6,7,8].

The goal of this review paper is to present a comprehensive overview of the most recent techniques for detecting glaucoma by means of deep learning methods applied to retinal fundus images. A discussion of the strengths and limitations of these techniques in addition to their potential to improve glaucoma screening accuracy and efficiency is also included in this paper. This review was based only on reliable academic databases, where recent articles were gathered from 2019 to 2023. We hope that this paper provides insight into the current state of glaucoma screening and helps identify gaps in the research and development in this area.

### 1.1. Information Sources

According to [5,9,10], an extensive search has been performed for glaucoma diagnosis-related articles by utilizing reputable databases, such as Scopus, ScienceDirect, IEEE Xplore, WoS, and PubMed Central. The compiled list comprises both medical and technical literature and provides a comprehensive representation of all research activities in this field.

### 1.2. Study Selection Procedures

To identify relevant papers, a two-stage approach was executed that involved screening and filtering processes. Both stages used the same criteria for including or excluding relevant papers. During the first stage, duplicate studies were removed as well as articles that were unrelated according to the preliminary review of the titles and abstracts. In the second stage, we thoroughly reviewed, analyzed, and summarized the remaining papers in order to obtain a set of relevant studies.

### 1.3. Search Mechanism

This study was conducted using a range of search keywords in highly reputable databases, such as IEEE Xplore, ScienceDirect, PubMed, Scopus, and WoS. Our search query comprised of two parts, linked with the operator “AND”. The first and second parts consist of different sets of keywords, with the former including “glaucoma” and “fundus images”, and “optic disc” and the latter comprising “deep learning”, “convolutional neural network”, and “CNN”. Within each part, the operator “OR” was used to connect the keywords. We only focused on scientific studies published in journals, and thus excluded conferences, books, and other forms of literature, while at the same time prioritizing only the current and relevant research on the use of deep learning in retinal disease, particularly glaucoma of all types including open-angle, angle-closure, and normal-tension glaucoma. Our research query and inclusion criteria are detailed in Figure 2. In addition, this work did not distinguish between patients who had previous treatment experience and those who did not. Both treatment-experienced and treatment-naïve patients were mixed up, since most of the works focused only on the resultant disease lesions.

Figure 3 shows the relationship between the number of articles collected from reliable databases in conjunction with the year of publication for each article. The collected articles included a presentation of the latest proposed deep learning techniques in the field of glaucoma diagnosis.

### 1.4. Paper Organization

The main objective of this review is to conduct a detailed analysis of various deep learning techniques recently used in the diagnosis of glaucoma diseases through the analysis of fundus images. In addition, it provides an extensive overview of the various datasets available for glaucoma disease, including their ground truth descriptions. This review additionally provides further details about deep learning frameworks, which are frequently employed in the diagnosis of retinal diseases, as well as widely used methodologies for image processing and evaluation metrics.

The review also takes into account a number of current research techniques that are related to this research focus. In brief, this review aims to provide readers with an in-depth and up-to-date understanding of the recent developments in the field of AI-based diagnosis of retinal diseases, especially with regard to glaucoma.

The organization of this article is as follows: Section 2 presents a brief summary of the various forms of glaucoma, the risk factors, the datasets readily accessible online for glaucoma diagnosis, and the evaluation metrics that are commonly utilized to measure the effectiveness of these models. Section 3 provides a breakdown of the different image pre-processing techniques that are very common for fundus imaging analysis. In Section 4, the most commonly used approaches for the detection of glaucoma are discussed along with specific backbone models that are used for both classification and segmentation tasks in fundus image analysis. Finally, Section 5 discusses several potential research gaps and the corresponding recommendations, as well as their limitations, and Section 6 concludes the article.

## 2. Glaucoma Overview: Types, Factors, and Datasets

Glaucoma, a complicated and progressive eye disease, has several kinds, phases, and risk factors. Understanding these categories and recognizing the risk factors is essential for proper diagnosis and treatment. Furthermore, the use of well-known public databases improves the reliability of diagnostic procedure outcomes and facilitates comparisons with related studies on this disease.

### 2.1. Types of Glaucoma

There are several types of glaucoma, which are open-angle glaucoma, angle-closure glaucoma, and normal-tension glaucoma [11,12]. Open-angle glaucoma, which is the most common type of glaucoma, occurs when the drainage angle in the eye becomes less efficient over time, leading to increased pressure inside the eye. Angle-closure glaucoma, on the other hand, occurs when the iris bulges outward and blocks the drainage angle, leading to a sudden increase in eye pressure [13]. Normal-tension glaucoma is a less common type that occurs when the optic nerve is damaged, even though the eye pressure is within the normal range. Treatment options for glaucoma include eye drops, medication, laser therapy, and surgery, and the choice of treatment depends on the type and severity of the disease [14].

### 2.2. Risk Factors for Glaucoma

There are several risk factors associated with glaucoma that include age, family history, high eye pressure, thin corneas, and certain medical conditions, such as diabetes and high blood pressure [15,16]. Other risk factors may include a history of eye injuries, long-term use of steroid medications, and a high degree of nearsightedness or farsightedness. Additionally, individuals of African, Hispanic, or Asian descent may be at higher risk of developing certain types of glaucoma. Figure 4 demonstrates the majority of glaucoma risk factors. Even though some risk factors are out of a patient’s control, regular eye tests and frequent early screening for glaucoma might be helpful in recognizing and treating this medical condition. Patients with risk factors must get their eyes checked out by an eye specialist on an ongoing schedule to check for glaucoma symptoms and hence prevent the likelihood of vision loss [17,18].

### 2.3. Retinal Fundus Image Datasets

For the purpose of diagnosing glaucoma, there are a wide variety of datasets that are freely available online that include datasets on retinal images, optical coherence tomography (OCT) scans, and individual clinical data with glaucoma condition [6,19]. These datasets often come with expert annotated labels, whereby the labels provide additional information on the exact location and severity of retinal damage resulting from glaucoma. Such information can be utilized as the ground truth for the development and evaluation of automated glaucoma screening tools, which needs to be verified by trained ophthalmologists or qualified experts in the field. Using these data, the development of novel methods for the early detection and monitoring of glaucoma has been facilitated. Consequently, public datasets have emerged as an essential resource for researchers, physicians, and other healthcare professionals aiming to enhance glaucoma diagnosis and treatment [20,21]. Table 1 shows some of the frequently employed public datasets for glaucoma diagnosis. Each of these datasets has distinct features that can be dedicated to particular applications or types of research topics as well. Typically, both healthy and glaucoma-affected images are included in the datasets, which is an important factor to enable the creation and assessment of algorithms that are capable of distinguishing between healthy and affected eyes.

Figure 5 shows the distribution of the available datasets and the number of times they were used by researchers for glaucoma diagnosis. These public datasets contain fundus images according to certain criteria with specific accuracy and dimensions, and they are often divided into two or more groups for the purpose of training, validation, and testing. In addition, some researchers use specialized datasets that are collected locally from hospitals specializing in the respective eye diseases. DRISHTI-GS1, RIM-ONE, and ORIGA are the most commonly used datasets among the reviewed articles.

## 3. Feature Enhancement and Evaluation

The purpose of image enhancement techniques is to increase the quality of images, which is an essential process for improving the analysis of images. Furthermore, ROI localization improves image analysis efficacy by detecting and extracting specific parts within an image that are relevant to a specific task. Finally, these operations, in addition to classification and/or segmentation operations, usually require evaluation measures that demonstrate the effectiveness of the systems applied to the data.

### 3.1. Pre-Processing Techniques

Retinal fundus imaging is a crucial modality used in diagnosing and monitoring various eye diseases. Despite advancements in retinal imaging technology, several challenges still exist, such as low image quality due to image artifacts, poor illumination, and motion blur [42,43].

In order to create accurate and reliable prediction models, it is common practice to pre-process fundus images prior to the training phase to minimize the effects of noise that can arise from the use of different image-capturing equipment in various illumination settings. Basically, deep learning networks often require fixed input dimensions for efficient processing. These networks commonly operate on fixed-size tensors as input, which means that the images need to be resized or transformed to a consistent size before being fed into the network.

Due to the complexity of the retinal structure, important biomarkers and lesions may not be easily identifiable in images of poor quality. In addition to noise reduction techniques, pre-processing techniques are utilized to enhance the important features of fundus images prior to the implementation of deep learning models [44,45,46]. Table 2 provides a list of commonly employed pre-processing methods for color fundus images in the diagnosis of retinal diseases.

### 3.2. Optic Nerve Head Localization

In most of the automated image processing operations that are employed for diagnostic purposes, the medical images are usually coupled with localization information that is helpful for diagnostic operations.

For fundus images, which are usually used to diagnose glaucoma, the locations of the optic disc or optic nerve head are crucial in diagnosing the disease. The optic disc is the entry point of the optic nerve, which transmits visual data to the brain. It is often called the “optic nerve head” with a circular shape area on the retina [47].

In general, the process of localizing this structure will help to identify abnormalities for diagnosing diseases later on. This process reduces the attention area size, which usually increases the diagnostic accuracy since only these specific areas are analyzed. However, this approach relies on the assumption that the rest of the areas are less important for diagnosing the disease.

To locate the optic disc or optic nerve head, there are several techniques that localize the center of the optic disc and calculate its radius to be used as input for the segmentation process [48]. For the optic cup, its localization information is less crucial as it is located within the optic disc and is often easily identified.

The following list describes some of the image processing techniques that are commonly used to perform localization of the optic disc and/or the optic cup [49]. These techniques include the following:Threshold: A certain boundary value is set to separate the optic disc and optic cup from the surrounding retina, depending on the density of image pixels;Edge Detection: Identify and detect edges of the optical disc and optical cup based on sudden changes in pixel values using algorithms such as the Sobel operator or Canny edge detector;Template Matching: Locating the optical disc or optic cup in the image using a binary template that matches their shapes;Machine Learning/Deep Learning: Training a network to identify the optic disc and optic cup in a fundus image based on a set of predefined features such as texture, size, and shape.

After successful localization of the optic disc and optic cup, this information will be used as an input to implement further image processing methods such as isolating these structures, determining their sizes, and identifying any irregularities that may be present.

### 3.3. Performance Metrics

Performance metrics are essential components used as an evaluation measure for deep learning applications to evaluate their accuracy and efficiency among the numerous proposed segmentation and classification methods [50]. For glaucoma diagnosis, the segmentation output of the retinal fundus image is crucial information used to detect the specific areas in which the disease is present.

The standard performance metrics that are involved in the segmentation task are Jaccard index, Dice similarity coefficient (DSC), and sensitivity, which are frequently utilized to assess the effectiveness of the segmentation process. These metrics measure how closely the segmentation result matches the actual disease-affected areas, namely, the ground truth [51,52].

For the classification task, the purpose of performance metrics is to measure how well the algorithm has correctly categorized the images as either having glaucoma or being in a healthy state. Hence, it is important to collect basic information on the number of correctly classified images. The performance of a classification task is typically evaluated using metrics such as the accuracy, precision, recall, F1 score, and area under the receiver operating characteristic curve (AUC-ROC).

Hence, holistic performance metrics are essential for assessing the performance of glaucoma diagnosis classification and segmentation applications. These metrics assist researchers in assessing the methods’ precision and figuring out how well they detect the presence of glaucoma.

The evaluation and comparison of various glaucoma diagnosis techniques depend heavily on the choice of appropriate performance metrics. Table 3 lists the most commonly used metrics along with their descriptions [47,53,54].

Depending on the extracted articles, different sets of evaluation matrices were employed, which are appropriate to the architecture of each network, whether for segmentation or classification tasks, which reveal the strength of the performance. Figure 6 shows the distribution of the most used evaluation matrices in the collected papers.

## 4. Glaucoma Detection

There are two primary tasks in which deep learning algorithms have played a vital role in diagnosing retinal-based diseases: classification and segmentation. The classification task involves the process of categorizing input images into different disease categories. On the other hand, the segmentation task involves the process of identifying critical lesions and significant biomarker areas from a given fundus image of a patient, which is used to discover more details about the type and nature of retinal diseases. Several deep learning models have been created and evaluated for these tasks [10,43,55]. Figure 7 depicts an overarching deep learning framework for diagnosing retinal disease.

### 4.1. Glaucoma Diagnosis

Glaucoma is a significant contributor to the irreversible loss of vision worldwide. Scientists have dedicated their efforts to developing diverse deep learning models aimed at identifying diseases from fundus images, similar to several other retinal ailments. Table 4 outlines a list of the experimental findings for the deep learning-based diagnosis of glaucoma.

Li et al. [26] proposed an “Attention-based AG-CNN” method that utilizes deep features highlighted by the visualized maps of pathological regions to automatically detect glaucoma. The guided back-propagation technique was used to locate small pathological regions based on predicted attention maps, enabling the refinement of attention maps to highlight the most important regions for glaucoma detection. This method has significant potential for automated glaucoma detection and identification of pathological regions in fundus images. Correspondingly, Wang et al. [56] trained deep models for glaucoma classification based on the transfer learning strategy of the VGG-16 and AlexNet architectures. In addition, images of the optic nerve head were customized from different publicly available datasets, which they divided into two subsets. One subset was augmented using various data augmentation techniques such as random scaling, shearing, rotation, and flipping, while the second subset of images was reconstructed by producing 3D topographical maps of the ONH using shading information from 2D images. After that, the generated datasets were evaluated for glaucoma classification and produced better performance than normal CNN classification models.

In another work, Gheisari et al. [57] introduced a new method for improving the accuracy of glaucoma classification by integrating the CNN frameworks of VGG16 and ResNet50 with the RNN-LSTM model. To enhance the classification performance, the proposed architecture combines static structural features with dynamic vessel features. The LSTM-based RNN was used because of its capacity to select and retain pertinent information from the image sequence in the hybrid module. To increase the accuracy of the hybrid module of glaucoma classification, a fully connected layer was added at the end of the network. On the other hand, Nayak et al. [58] suggested a new type of network architecture called “ECNet” based on fundus images for effective glaucoma detection. Convolutional, compression, ReLU, and summation layers form the basic structure of the model to extract the significant disease features. A real-coded genetic algorithm (RCGA) was used to optimize the learnable layer weights, as opposed to gradient-based algorithms, which may lead to an overfitting problem, and thus, the need for a larger training dataset. A variety of classifiers were used for the classification task, with RCGA and SVM producing the best outcomes.

Additionally, Li et al. [59] proposed a deep learning approach for identifying glaucomatous damage in fundus images using pre-trained ResNet101. To address the vanishing gradient problem during training, skip connections between the layers were utilized. These connections perform identity mapping without adding additional parameters or computational complexity. The authors also explored the integration of short medical history data with fundus images, which resulted in a slight performance improvement in the model. Furthermore, Hemelings et al. [60] proposed a novel model for glaucoma detection using deep learning and active learning techniques. The model incorporates transfer learning from ResNet-50 architecture and utilizes pre-processing techniques such as ROI extraction and data augmentation. An active learning strategy through uncertainty sampling was utilized to leverage uncertainty information from an unlabeled dataset to reduce the labeling cost. Furthermore, the model generates interpretable heat maps to support decision-making. For the same reason, Juneja et al. [61] introduced a new method called “GC-NET” for glaucoma classification using retinal fundus images. Their approach involved three pre-processing techniques, namely, image cropping, augmentation, and denoising, to eliminate irrelevant details from the input images. They then utilized a 76-layer deep CNN-based model, which included an ‘Add layer’ in every block to minimize data loss by combining the previous block output with the next block output, except for the first and final blocks.

On the one hand, Liu et al. [62] proposed a DL framework, “GD-CNN” for automatic diagnosis of glaucoma using fundus images that was trained on a large dataset of positive and negative cases from different sources. The network is based on the ResNet model and the stochastic gradient descent optimizer for binary classification. To improve the generalization ability of the model, an online DL system was introduced in which ophthalmologists iteratively confirmed the model’s classification results, whereby the confirmed samples were used for fine-tuning before making new predictions. Moreover, Bajwa et al. [63] proposed a two-stage cascade model for glaucoma screening. In the first stage, a heuristic method based on regions with CNN (RCNN) was used for the extraction and localization of OD from a retinal fundus image. This stage includes a sub-model using a rule-based algorithm to generate semi-automated annotations for OD, used during the training of the RCNN. In the second stage, a deep CNN was used to classify the extracted ROI images into two classes: glaucoma and non-glaucoma.

Apart from this, Kim et al. [64] proposed a two-pronged approach for glaucoma classification and localization. The first approach involves the glaucoma classification phase using three different CNN architectures (VGGNet, InceptionNet, and ResNet) with a variety of regularization techniques being used to increase the model’s generalization. The ResNet-152-M network produced the best diagnostic results. The second approach utilizes localization detection based on a weekly supervised method called “Grad-CAM” to identify glaucoma regions in an input image without using any segmentation data. This work also involved the development of a web-based application for locating and diagnosing glaucoma in a limited medical setting. In another work, Hung et al. [65] proposed the use of a pre-trained Efficient-Net-b0 as a base and incorporated additional patient features such as age, gender, and high myopia for binary and ternary classification of glaucoma. The binary classification sub-model task is to distinguish between glaucoma and the non-glaucoma optic disc, whereas the ternary sub-model aims to classify input images into a healthy optic disc, high-tension glaucoma, or normal-tension glaucoma. To avoid the possibility of overfitting as well as increasing the model performance, pre-processing techniques are applied first to the input images.

However, Cho et al. [66] proposed ternary classification framework of glaucoma stages based on an ensemble deep learning system. The system consists of 56 sub-models created by combining two types of fundus images, seven image filters, and four CNN architectures. A set of pre-processing techniques is first applied to the input images that include data augmentation, resizing, and filtering. Based on the average probabilities of all sub-models, the final classification decision is made, and the class with the highest probability is chosen to be the output. Leonardo et al. [67] proposed the integration of generative modeling and deep learning to improve the diagnostic performance of glaucoma by converting low-quality images into better-quality images using U-Net-based generative adversarial networks. Furthermore, the quality of the generated images was evaluated using a pre-trained CNN (EfficientNetB0), where low-quality images were excluded, while high-quality images were preserved. In addition, Alghamdi et al. [68] proposed employing three CNN architectures based on transfer, supervised, and semi-supervised learning techniques, where the three models were trained on public databases of fundus images without applying any pre-processing or enhancement techniques to the selected data. First, the transfer learning algorithm was applied to a limited dataset, and then semi-supervised learning was applied to the labeled and unlabeled datasets. Finally, unsupervised learning was applied with the supervised learning stage by using a 6-layer CNN autoencoder model to extract the necessary features.

In comparison, Devecioglu et al. [40] employed heterogeneous “Self-Organized Operational Neural Networks (Self-ONNs)” as an alternative diagnosis system to address dataset limitations and reduce the computing burden. For the purpose of evaluating the quality of the proposed model in comparison to several other trained CNN models, it was found that the proposed model produced superior performance compared to the benchmarked models. Furthermore, Juneja et al. [69] proposed a classification system based on a modified Xception network to extract precise features from the optic disc and optic cup located in the center of the retinal fundus image using fewer layers and larger filter sizes that enable self-learning for diagnosing glaucoma at the early stages. In addition, the input images are cropped and augmented to reduce the image size and computational time in order to improve the performance of the proposed model.

Carvalho et al. [70] have proposed an automated system for diagnosing glaucoma using retinal imaging, which is based on an adapted three-dimensional convolutional neural network (3DCNN). In contrast to other methods, the proposed system does not require optic disc segmentation masks. The system converts each two-dimensional input image into four volumes that represent the red, green, blue, and gray levels using a specialized technique that deeply extracts low-level features. The conventional VGG16 architecture is modified to process the generated volumes, and it is found that a better performance is achieved by increasing the number of layers. The gray-level images exhibit superior results for glaucoma classification compared to the RGB channels. Moreover, Joshi et al. [41] have presented a method for the early diagnosis of glaucoma using an ensemble of three pre-trained convolutional neural network architectures based on transfer learning: ResNet50, VGG16Net, and GoogLeNet. The proposed method employs pre-processed retinal fundus images from both public and private databases to extract features using convolutional neural networks. The extracted features then become the input to a classifier to categorize the images as normal or abnormal using the maximum voting technique.

Apart from this, Almansour et al. [71] have designed a comprehensive deep learning framework for the early detection of glaucoma by identifying morphological symptoms “(peripapillary atrophy)” in retinal fundus images. The detection methodology was executed through two deep learning models that operate sequentially on both public and local dataset images. The first model uses a mask region-based CNN (R-CNN) to localize and crop the region of interest, which acts as a pre-processing technique to enhance the performance of the framework. The second model employs three pre-trained CNN algorithms, VGG-16, ResNet-50, and Inception-V3, to classify the presence of symptoms (i.e., detect glaucoma) in the cropped regions. Aamir et al. [72] proposed a framework for the detection and classification of glaucoma using retinal fundus images based on a multi-level deep convolutional neural network “(ML-DCNN)” architecture. The proposed framework consists of three stages. In the first stage, a pre-processing step is performed using an adaptive histogram equalization technique to reduce image noise. The pre-processed images are then fed into the second stage, where a CNN detection network “DN-CNN” based on feature learning is used to detect glaucoma. Finally, the detected samples are classified into three statistical levels (Early, Moderate, and Advanced) using a CNN classification network that is dedicated to each level.

For the same reason, Islam et al. [73] proposed a classification framework for glaucoma diagnosis using two distinct datasets derived from the cropped OD/OC in addition to the segmented blood vessel dataset built using the U-Net architecture. The datasets were pre-processed with multiple augmentation methods to increase the dataset size and perform generalization. After that, four deep learning algorithms (EfficientNet, MobileNet, DenseNet, and GoogLeNet) were applied to classify the images into two categories. The EfficientNet b3 model offers the best performance for cropped fundus images. Moreover, Liao et al. [74] proposed a deep learning framework for glaucoma detection that highlights distinct areas identified by the network, providing a clinically interpretable view of the disease. The framework includes a ResNet backbone network for feature extraction, multi-layer average pooling to bridge the gap in information at different scales, and evidence activation mapping for reliable diagnosis.

On the one hand, Sudhan et al. [75] have put forth a framework that is based on deep learning techniques to achieve the tasks of segmentation and classification of glaucoma. In their proposed approach, the commonly used U-Net architecture is employed for the purpose of segmentation. Once segmentation is completed, a pre-trained DenseNet-201 architecture is employed to extract the features from the segmented images. Subsequently, a DCNN (Deep Convolutional Neural Network) architecture is utilized to classify the images and identify glaucoma cases. Furthermore, Nawaz et al. [76] employed EfficientNet-B0 to extract deep features from fundus images. These features were then fed into the “Bi-directional Feature Pyramid Network” module of (EfficientDet-D0), which applies bi-directional key point fusion iteratively on the extracted features. The resulting localized area that contains the glaucoma lesion was then used to predict the presence of glaucoma.

In another study, Diaz-Pinto et al. [77] fine-tuned five deep learning models (VGG16, VGG19, InceptionV3, ResNet50, and Xception) for glaucoma classification after pre-training the models on ImageNet. The last fully connected layer of each model was replaced with a global average pooling layer, followed by a fully connected layer with two nodes and a SoftMax classifier. Their findings concluded that Xception achieved the best performance. In the same way, Serte et al. [78] aimed to develop a glaucoma detection model that uses graph-based saliency to crop the optic disc and remove irrelevant portions of fundus images. The model then locates the optic disc and feeds it into a set of three powerful CNN architectures: namely, AlexNet, ResNet-50, and ResNet-152.

Moreover, Jos’e Martinsa et al. [79] proposed a U-shaped architecture for jointly segmenting the optic disc and optic cup, comprising four depth levels in the encoding path with two depth-wise separable convolution blocks in each level. To enhance the spatial context in the decoding path, skip connections were added at every depth level. The encoder-to-decoder path transition was augmented with an ASPP module featuring four parallel padded atrous convolutions. Furthermore, a classification network utilizing a MobileNetV2 model as a feature extractor was developed to generate a confidence level classifier to detect glaucoma. Correspondingly, Natarajan et al. [7] introduced “UNet-SNet,” which is a new and effective two-stage framework for achieving high segmentation and classification accuracies. In the first stage, a regularized U-Net is utilized as a semantic segmentation network to perform optic disc segmentation. In the second stage, a glaucoma detection model is implemented using a fine-tuned SqueezeNet CNN.

**Table 4 diagnostics-13-02180-t004:** Glaucoma diagnosis performance comparison.

Reference	Dataset	Camera	ACC	SEN	SPE	AUC	F1
Li et al. [26]	Private—LAG	Topcon, Canon, Zeiss	0.962	0.954	0.967	0.983	0.954
	RIM-ONE		0.852	0.848	0.855	0.916	0.837
Wang et al. [56]	DRIONS-DB, HRF-dataset, RIM-ONE, and DRISHTI-GS1	-	0.943	0.907	0.979	0.991	-
Gheisari et al. [57]	Private	Carl Zeiss	-	0.950	0.960	0.990	0.961
Nayak et al. [58]	Private	Zeiss FF 450	0.980	0.975	0.988	-	0.983
Li et al. [59]	Private	Zeiss Visucam 500, Canon CR2	0.953	0.962	0.939	0.994	-
Hemelings et al. [60]	Private	Zeiss Visucam	-	0.980	0.910	0.995	-
Juneja et al. [61]	DRISHTI-GS and RIM-ONE	-	0.975	0.988	0.962	-	-
Liu et al. [62]	Private	Topcon, Canon, Carl Zeiss	-	0.962	0.977	0.996	-
Bajwa et al. [63]	ORIGA, HRF, and OCT & CFI	-	-	0.712	-	0.874	-
Kim et al. [64]	Private	-	0.960	0.960	1.000	0.990	-
Hung et al. [65]	Private	Zeiss Visucam, Canon CR-2AF, and KOWA	0.910	0.860	0.940	0.980	0.860
Cho et al. [66]	Private	Nonmyd7, KOWA	0.881	-	-	0.975	-
Leonardo et al. [67]	ORIGA, DRISHTI-GS, REFUGE, RIM-ONE (r1, r2, r3), and ACRIMA	-	0.931	0.883	0.957	-	-
Alghamdi et al. [68]	RIM-ONE and RIGA	-	0.938	0.989	0.905	-	-
Devecioglu et al. [40]	ACRIMA	-	0.945	0.945	0.924	-	0.939
	RIM-ONE	-	0.753	0.682	0.827	-	0.739
	ESOGU	-	1.000	1.000	1.000	-	1.000
Juneja et al. [69]	-	-	0.935	0.950	0.990	0.990	-
De Sales et al. [70]	DRISHTI-GS and RIM-ONEv2	-	0.964	1.000	0.930	0.965	-
Joshi et al. [41]	DRISHTI-GS, HRF, DRIONS-DB, and one privet dataset PSGIMSR	-	0.890	0.813	0.955	-	0.871
Almansour et al. [71]	(RIGA, HRF, Kaggle, ORIGA, and Eyepacs) and one privet dataset (KAIMRC)	-	0.780	-	-	0.870	-
Aamir et al. [72]	Private	-	0.994	0.970	0.990	0.982	-
Liao et al. [74]	ORIGA	-	-	-	-	0.880	-
Sudhan et al. [75]	ORIGA	-	0.969	0.970	0.963	-	0.963
Nawaz et al. [76]	ORIGA	-	0.972	0.970	-	0.979	-
Diaz-Pinto et al. [77]	ACRIMA, DRISHTIGS1, sjchoi86-HRF, RIM-ONE, HRF	-	-	0.934	0.858	0.960	-
Serte et al. [78]	HARVARD	-	0.880	0.860	0.900	0.940	-
Jos’e et al. [79]	ORIGA, DRISHTI-GS, RIM-ONE-r1, RIM-ONE-r2, RIM-ONE-r3, iChallenge, and RIGA	-	0.870	0.850	0.880	0.930	-
Natarajan et al. [7]	ACRIMA	-	0.999	1.000	0.998	1.000	0.998
	Drishti- GS1	-	0.971	1.000	0.903	0.999	0.979
	RIM-ONEv1	-	1.000	1.000	1.000	1.000	1.000
	RIM-ONEv2	-	0.999	0.990	0.995	0.998	0.992
Islam et al. [73]	HRF and ACRIMA	-	0.990	1.000	0.978	0.989	-

Table 5 shows the summary of the review and investigation of the glaucoma diagnosis systems developed in the selected studies.

### 4.2. Optic Disc/Optic Cup Segmentation

Several essential biomarkers utilized in diagnosing glaucoma include the Optic Disc and Optic Cup. The cup-to-disc ratio is obtained by calculating the ratio of the diameter of vertical cup diameter over the vertical disc diameter. Consequently, precise segmentation of OD/OC has turned out to be crucial for glaucoma diagnosis, and a considerable amount of research has been conducted in this regard. Recent studies centered on deep learning-based segmentation of OD/OC are examined in the subsequent sections, and their corresponding experimental outcomes are presented in Table 6.

Civit-Masot et al. [80] have proposed a diagnostic aid tool for glaucoma that employs two independent sub-models to generate a diagnosis report for ophthalmologists. The first sub-model utilized two generalized U-Net architectures to segment the optic disc and optic cup independently by extracting their physical and positional characteristics. The second sub-model used a pre-trained MobileNet-V2 architecture to directly classify the fundus image without applying any segmentation network. The outputs of both sub-models were combined to create a comprehensive report that was used to assist ophthalmologists in the diagnosis process. In a similar manner, Pascal et al. [81] developed a multi-task DL model for glaucoma diagnosis that uses a modified U-Net architecture with a pre-trained VGG-16 backbone. The model’s goal is to segment the OD and OC, to localize the fovea, and to detect glaucoma based on retinal fundus images, all through a specialized optimization scheme and additional skip connections between the encoder and decoder layers.

Apart from that, Shanmugam et al. [82] proposed a glaucoma recognition model that uses deep learning-based segmentation to estimate the cup-to-disc ratio and fed the resultant segmentation mask to a random forest classifier to classify fundus images into either glaucoma or normal categories. Their model employs a modified U-Net called “Au-Net” to segment the OD and OC, which are used to estimate the CDR values. Moreover, Cheng et al. [83] developed a “Disc-aware Ensemble Network (DENet)” for glaucoma screening. Their proposed network represents fundus image information at both global and local levels, combining four sub-models: ResNet for direct image classification, U-Net for disc area segmentation, probability screening from the segmented disc area, and polar transformation to improve segmentation accuracy. These four outputs were combined to provide the final screening result.

Furthermore, Sreng et al. [84] proposed a two-stage deep learning framework for glaucoma screening using a combination of various convolutional neural networks. The first stage segmented the OD area using a modified DeepLabv3+ architecture based on five network configurations. The second stage used eleven pre-trained CNNs to extract the OD area features utilizing three ways: transfer learning, support vector machine, or a hybrid of transfer learning and support vector machine. On the one hand, Yu et al. [85] proposed a glaucoma detection model that uses a modified U-Net architecture with pre-trained ResNet34 as the backbone for segmenting the OD and OC. The segmentation process occurs in two stages, a conventional U-Net is first used to segment the ROI before a modified U-Net is used to obtain more accurate segmentation outputs for the OD and OC in the second stage. Additional post-processing techniques were also applied to calculate the vertical diameter CDR.

Similarly, Natarajan et al. [86] proposed a glaucoma detection framework using a combination of pre-processing, segmentation, and classification techniques. The images were pre-processed using CLAHE and then segmented into super-pixels (ROI) using only the green channel information. The modified kernel fuzzy C-means algorithm was applied for accurate detection of the OD and OC. A set of GLCM features was extracted from the last segmentation stage, which were then fed to the VGG16 classifier to determine the stage of the glaucoma disease. Moreover, Ganesh et al. [87] developed a DL model called “GD-Ynet” for optic disc segmentation and binary glaucoma classification. The model is based on a modified U-Net architecture that uses inception modules instead of basic convolutional layers to extract low-level features. The proposed model uses contextual features of activation maps to capture the ROI and perform optic disc segmentation. Then, aggregated transformations are used to perform binary classification for glaucoma detection from the detected optic disc.

Furthermore, Juneja et al. [88] proposed the Glaucoma Network (G-Net), which is a deep convolutional neural network framework for glaucoma diagnosis from retinal fundus images. G-Net also used a modified U-Net architecture with two separate CNNs working together to segment the (OD) and (OC), which is then used as the input to calculate the CDR. The input images are pre-processed and cropped, and only the red channel is utilized for OD segmentation, while all three channels of RGB are used for OC segmentation. The resultant data is augmented before it is fed to G-Net to detect glaucoma using the calculated CDR. For the same reason, Veena et al. [89] proposed a framework for glaucoma diagnosis also using two separate CNN models to accurately segment the optic cup and optic disc; these two pieces of information are also used to determine the CDR ratio. The models each have 39 layers of CNN to extract a greater set of features with the aim of reducing feature inconsistency. The images are pre-processed using morphological operations to improve the contrast level, Sobel filter for feature extraction, and Watershed algorithm for optic nerve localization. The resultant output is then inputted to both models to calculate the CDR.

In another work, Tabassum et al. [90] proposed a “Dense Cup Disc Encoder-Decoder Network” to segment the OD and OC without performing any localization or pre-/post-processing methods. The encoder facilitates feature reuse, and the decoder allows information reuse, thereby reducing the need for feature upsampling and lowering the number of network parameters without sacrificing the performance. Moreover, Liu et al. [91] proposed a deep separable convolution network with a dense connection as its core, complemented by a multi-scale image pyramid to enhance the network capacity. Image morphology was also used for post-processing the segmentation outcomes, and the optic disc’s center was localized using a CNN and Hough circle detection. A high-precision segmentation network was then trained using the extracted region of interest to accurately segment the optic disc and cup.

Furthermore, Nazir et al. [92] proposed a deep learning approach for the automated segmentation of the optic disc and optic cup from retinal images using a customized Mask-RCNN model. They applied a data augmentation technique by adding blurriness variations to increase the data diversity and generate the ground truth annotations. The authors incorporated the “DenseNet-77” model at the feature computation level of Mask-RCNN to compute a broader range of key points, enabling more precise localization of the OD and OC regions across various sample conditions. In addition, Rakhshanda et al. [93] employed a pixel-wise semantic segmentation model to identify the optic disc and optic cup using an encoder–decoder network. Augmented data are combined with the existing training data, which are then processed by a VGG-16 network to generate a set of feature vectors for OD, OC, and background classification. The segmentation outcomes are then utilized to calculate the CDR, which assists in the diagnosis and analysis of glaucoma.

On the one hand, Wang et al. [94] proposed an asymmetrical segmentation network that combines U-Net with a novel cross-connection subnetwork and decoding convolutional block for OD segmentation. The network also employs multi-scale input features to mitigate the impact of consecutive pooling operations. The integration of these features enhances the network’s ability to detect morphological variations in the respective regions-of-interest, while minimizing the loss of important features in the images. Similarly, Kumar et al. [95] proposed a novel approach for generating precise and accurate ground truth data by incorporating morphological operations. The U-Net architecture of 19 CNN layers with encoder and decoder blocks is utilized to discern spatial features. As a result, the model managed to improve the prediction performance of the optic disc region with greater precision and accuracy.

For the same reason, Panda et al. [96] proposed a glaucoma diagnosis deep learning approach for segmenting the optic disc and optic cup in fundus images with a limited number of training samples. This approach employs post-processing techniques, residual learning with skip connections, patch-based training, and other techniques to produce smoother boundaries and an even more accurate cup-to-disc ratio. Furthermore, Fu et al. [97] proposed a data-driven deep learning technique that employs the use of a U-Net architecture to segment the optic disc in abnormal retinal fundus images. The method employs the use of model-driven probability bubbles to determine the precise position of the optic disc and eliminate interference from light lesions, which eventually improves segmentation accuracy. Similarly, Zhao et al. [98] introduced a simplified approach that improves the accuracy of segmentation of fundus images, reduces the number of parameters, and reduces processing time by employing attention U-Net architecture and transfer learning, whereby the attention gate is placed between the encoder and decoder to put more emphasize on selected target regions.

Another attention-based network was suggested by Hu et al. [99] through an encoder–decoder-based segmentation network that includes a multi-scale weight-shared attention module and a densely linked depth-wise separable convolution module to address the issues brought on by differences in acquisition devices. The multi-scale weight-shared attention module, which is located at the top layer of the encoder, integrates both multi-scale OD and OC feature information using both channel and spatial attention processes. The densely connected depth-wise separable convolution module is integrated as the output layer of the network. Moreover, Baixin et al. [100] introduced a semantic segmentation model named the “Aggregation Channel Attention Network”, which relies heavily on contextual data. The model uses an encoder–decoder framework where a pre-trained DenseNet sub-model is included in the encoding layer and feature information from various resolutions is included in the decoding layer, which is subsequently integrated with an attention mechanism. The network can maintain spatial information using high-level characteristics to direct low-level features. To further improve network efficiency, the classification framework is also strengthened by means of cross-entropy information.

Furthermore, Shankaranarayana et al. [101] introduced a deep learning model for estimating retinal depth from a single fundus image by employing a fully convolutional network topology with a dilated residual inception block to perform multiscale feature extraction. A new pre-trained strategy called pseudo-depth reconstruction was proposed to take control over the problem of insufficient data for depth estimation. This study made another contribution by introducing a fully convolutional guided network to perform semantic segmentation based on a multi-modal fusion block that extracts the features from two separate modalities. Furthermore, Bengani et al. [102] proposed an encoder–decoder deep learning model that employs semi-supervised and transfer learning techniques to segment the optic disc in retinal fundus images. In order to extract features from unlabeled images, an autoencoder reconstructs the input images and applies a network constraint. The transfer learning technique is used to transform the pre-trained model into a segmentation network, where it is fine-tuned using ground truth labels. In the same way, Wang et al. [103] presented a patch-based output space adversarial learning context that can perform the segmentation for the optic disc and optic cup simultaneously. The approach employs a lightweight segmentation network and unsupervised domain adaptation to address domain shift challenges. The framework uses a patch-based approach for fine-grained discrimination of local segmentation details. The segmentation network combines the designs of DeepLabv3+ and MobileNetV2 to extract multi-scale discriminative context features while minimizing the computational burden.

**Table 6 diagnostics-13-02180-t006:** Segmentation performance comparison of the reviewed related papers.

Reference	Dataset	OD/OC	ACC	SEN	SPE	PRE	AUC	IoU/Jacc	F1	DSC	δ
Civit-Masot et al. [80]	DRISHTI-GS	OD	0.880	0.910	0.860	-	0.960	-	-	0.930	-
		OC	-	-	-	-	-	-	-	0.890	-
	RIM-ONEv3	OD	-	-	-	-	-	-	-	0.920	-
		OC	-	-	-	-	-	-	-	0.840	-
Pascal et al. [81]	REFUGE	OD	-	-	-	-	0.967	-	-	0.952	-
		OC	-	-	-	-	-	-	-	0.875	-
Shanmugam et al. [82]	DRISHTI-GS	OD	0.990	0.870	0.920	-	-	-	-	-	-
		OC	0.990	0.860	0.950	-	-	-	-	-	-
Fu et al. [83]	SCES	OD	0.843	0.848	0.838	-	0.918	-	-	-	-
	SINDI		0.750	0.788	0.712	-	0.817	-	-	-	-
	REFUGE	OD	0.955	-	-	-	0.951	-	-	-	-
Sreng et al. [84]	ACRIMA		0.995	-	-	-	0.999	-	-	-	-
	ORIGA		0.90	-	-	-	0.92	-	-	-	-
	RIM–ONE		0.973	-	-	-	1	-	-	-	-
	DRISHTI–GS1		0.868	-	-	-	0.916	-	-	-	-
Yu et al. [85]	RIM–ONE	OD	-	-	-	-	-	0.926	-	0.961	-
		OC	-	-	-	-	-	0.743	-	0.845	-
	DRISHTI–GS1	OD	-	-	-	-	-	0.949	-	0.974	-
		OC	-	-	-	-	-	0.804	-	0.888	-
Natarajan et al. [86]	DRIONS	OD/OC	0.947	0.956	0.904	0.997	-	-	-	-	-
Ganesh et al. [87]	DRISHTI-GS	OD	0.998	0.981	0.980			0.997	-	0.995	-
Juneja et al. [88]	DRISHTI-GS	OD	0.959	-	-	-	-	0.906	0.935	0.950	-
		OC	0.947	-	-	-	-	0.880	0.916	0.936	-
Veena et al. [89]	DRISHTI -GS	OD	0.985	-	-	-	-	0.932	0.954	0.987	-
		OC	0.973	-	-	-	-	0.921	0.954	0.971	-
Tabassum et al. [90]	DRISHTI-GS	OD	0.997	0.975	0.997	-	0.969	0.918	-	0.959	-
		OC	0.997	0.957	0.998	-	0.957	0.863	-	0.924	-
	RIM-ONE	OD	0.996	0.973	0.997	-	0.987	0.910	-	0.958	-
		OC	0.996	0.952	0.998	-	0.909	0.753	-	0.862	-
Liu et al. [91]	DRISHTI-GS	OD	-	0.978	-	0.978	-	0.957	-	0.978	-
		OC	-	0.922	-	0.915	-	0.844	-	0.912	-
	REFUGE	OD	-	0.981	-	0.941	-	0.924	-	0.960	-
		OC	-	0.921	-	0.875	-	0.807	-	0.890	-
Nazir et al. [92]	ORIGA	OD	0.979	-	-	0.959	-	0.981	0.953	-	-
		OC	0.951	-	-	0.971	-	0.963	0.970	-	-
Rakhshanda et al. [93]	DRISHTI–GS1	OD	0.997	0.965	0.998	-	0.996	-	-	0.949	-
		OC	0.996	0.944	0.997	-	0.957	-	-	0.860	-
Wang et al. [94]	MESSIDOR	OD	-	0.983	-	-	-	0.969	-	0.984	-
	ORIGA		-	0.990	-	-	-	0.960	-	0.980	-
	REFUGE		-	0.965	-	-	-	0.942	-	0.969	-
Kumar et al. [95]	DRIONS-DB	OD	0.997	-	-	-	-	0.983	-	-	-
	RIM-ONE		-	-	-	-	-	0.979	-	-	-
	IDRiD		-	-	-	-	-	0.976	-	-	-
Panda et al. [96]	RIM-ONE	OD	-	-	-	-	-	-	-	0.950	-
		OC	-	-	-	-	-	-	-	0.851	-
	ORIGA	OD	-	-	-	-	-	-	-	0.938	-
		OC	-	-	-	-	-	-	-	0.889	-
	DRISHTI–GS1	OD	-	-	-	-	-	-	-	0.953	-
		OC	-	-	-	-	-	-	-	0.900	-
Fu et al. [97]	(DRIVE, Kaggle, MESSIDOR, and NIVE)	OD	-	-	-	-	0.991	-	-	-	-
			-	-	-	-	-	-	-	-	-
			-	-	-	-	-	-	-	-	-
			-	-	-	-	-	-	-	-	-
X. Zhao et al. [98]	DRISHTI-GS	OD	0.998	0.949	0.999	-	-	0.930	-	0.964	-
		OC	0.995	0.877	0.998	-	-	0.785	-	0.879	-
	RIM-ONEv3	OD	0.996	0.924	0.999	-	-	0.887	-	0.940	-
		OC	0.997	0.813	0.999	-	-	0.724	-	0.840	-
Hu et al. [99]	RIM-ONE-r3	OD	-	-	-	-	-	0.917	-	0.956	-
		OC	-	-	-	-	-	0.724	-	0.824	-
	REFUGE	OD	-	-	-	-	-	0.931	-	0.964	-
		OC	-	-	-	-	-	0.813	-	0.894	-
	DRISHTI-GS	OD	-	-	-	-	-	0.950	-	0.974	-
		OC	-	-	-	-	-	0.834	-	0.900	-
	MESSIDOR	OD	-	-	-	-	-	0.944	-	0.970	-
	IDRiD	OD	-	-	-	-	-	0.931	-	0.964	-
Shankaranarayana et al. [101]	ORIGA	OD/OC	-	-	-	-	-	-	-	-	0.067
	RIMONE r3	OD/OC	-	-	-	-	-	-	-	-	0.066
	DRISHTI–GS1	OD/OC	-	-	-	-	-	-	-	-	0.105
Bengani et al. [102]	DRISHTI GS1	OD	0.996	0.954	0.999	-	-	0.931	-	0.967	-
	RIM-ONE	OD	0.995	0.873	0.998	-	-	0.882	-	0.902	-
Wang et al. [103]	DRISHTI-GS	OD/OC	-	-	-	-	-	-	-	-	0.082
	RIM-ONE-r3	OD/OC	-	-	-	-	-	-	-	-	0.081

Table 7 shows the summary of the review and investigation of the OD/OC segmentation systems developed in the selected studies.

## 5. Research Gaps, Recommendations and Limitations

### 5.1. Research Gaps

Glaucoma detection is a crucial task in the field of ophthalmology, as early detection can prevent blindness. In general, the reviewed articles successfully designed an automated system to diagnose glaucoma using retinal images. However, there are still some research gaps in the existing systems, as illustrated in Table 8.

### 5.2. Future Recommendations

To analyze the retina, researchers normally follow several steps that depend on the research objective and the proposed techniques. The researchers may include some objectives that include enhancement, localization, segmentation, or classification. Enhancement, localization, and segmentation tasks are usually implicit steps in a classification module. For a glaucoma diagnosis system, feature extraction and classification are the most necessary steps, while the other steps can be optional depending on the proposed algorithm. Additionally, researchers often explore modifications and variations of the architectures used to further enhance the performance of glaucoma diagnosis. ResNet and GoogLeNet have shown promising performance for glaucoma diagnosis; however, the specific architecture choice may depend on the dataset and research context. The fundamental processes and essential observations for glaucoma diagnosis are outlined as follows:Pre-processing is crucial for effective analysis;Annotating a diverse set of labels is more important than having a large quantity of annotations;Performance can be improved through fine-tuning and augmentation techniques;Complex features can be captured by applying deeper neural networks;Sufficient training data is critical for producing a high-accuracy system;Additional loss functions can be integrated to prevent overfitting in specific domains;Multi-scale CNN can also provide better feature extraction through various scale strategies;Medical expertise is valuable for understanding the underlying structure of diseases.

### 5.3. Limitations of the Study

The reviewed studies have a common limitation due to the same source of development databases. Nevertheless, the reviewed works are fairly diverse and are representative samples of the chosen sources. However, an intelligent eye disease screening and diagnosing system for various diseases should be integrated, not limited to glaucoma only. Furthermore, analyzing research activities that use deep learning methods for the diagnosis of these critical retinal diseases may not necessarily reflect the views and responses of the broader research community.

## 6. Conclusions

In the field of healthcare, digital image processing and computer vision methods are employed for disease screening and diagnostic purposes. Among the various eye disorders, glaucoma is one of the chronic conditions that can result in irreversible loss of vision because of damage to the optic nerve. Color fundus imaging is a good imaging modality for medical image analysis, whereby the deep learning models have been extensively researched for the automated diagnosis systems. This review presents a process-based approach to deep learning in glaucoma diagnosis that discusses publicly available datasets and their ground truth descriptions. Some datasets comprise high-quality images taken in a controlled environment, while others have images captured in diverse environmental conditions, which can steer deep model behavior towards practical applications. Combining datasets can be used to train a robust model for real clinical implementation. Pre-processing techniques, such as image augmentation and filtering, are generally able to improve disease-relevant feature extraction. Different backbones of deep models have been explored for classification and segmentation tasks that also include different learning paradigms, such as ensemble and transfer learning techniques, in order to improve the proposed model performance. The deep learning approach has shown good performance for retinal disease diagnosis and has even surpassed expert performance in some cases. However, integrating DL models into clinical practice remains a big future challenge due to limited data, interpretability needs, validation requirements, and trust-building. To tackle this, efforts have focused on improving data quality, enhancing interpretability, establishing protocols, and addressing ethical concerns through transparency and bias mitigation.

## Figures and Tables

**Figure 1 diagnostics-13-02180-f001:**
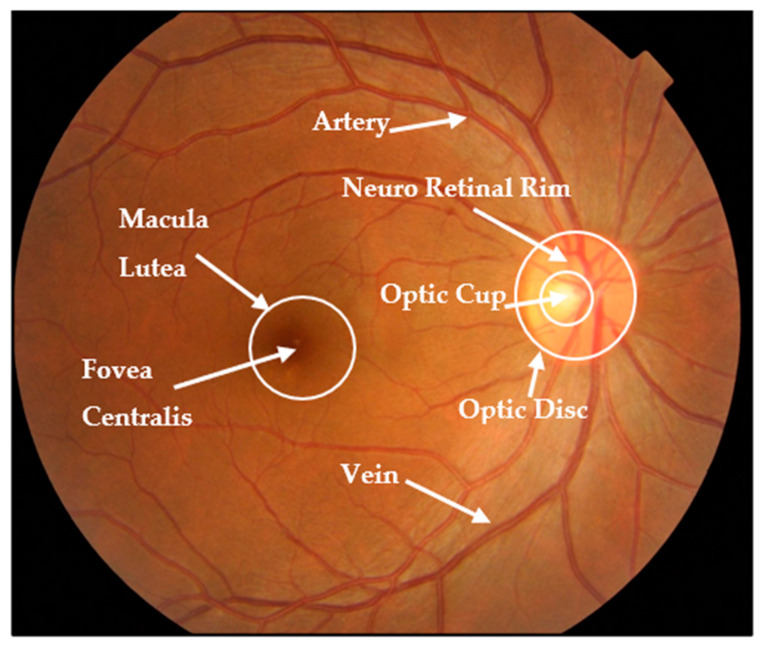
Fundus image structure.

**Figure 2 diagnostics-13-02180-f002:**
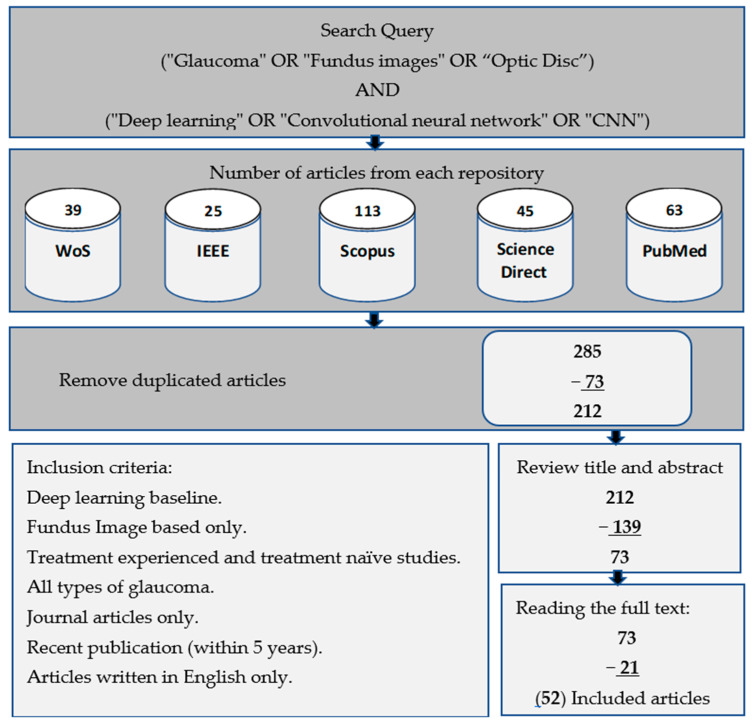
Workflow process for selecting relevant studies based on research queries and inclusion criteria.

**Figure 3 diagnostics-13-02180-f003:**
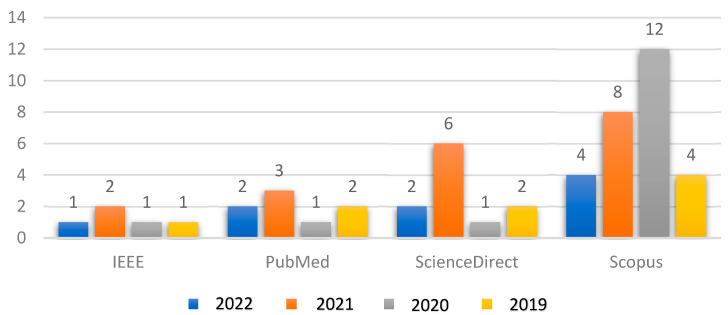
Relationship between the collected articles with respect to the years of publication.

**Figure 4 diagnostics-13-02180-f004:**
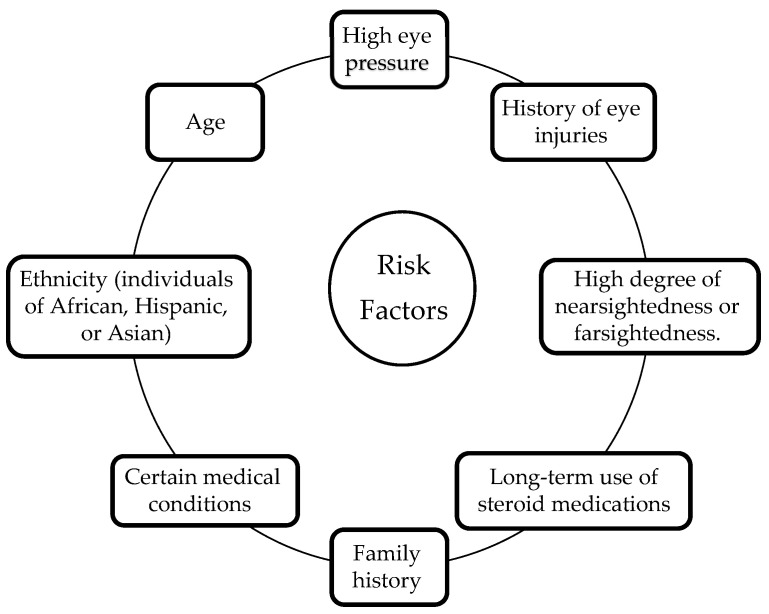
Glaucoma risk factors.

**Figure 5 diagnostics-13-02180-f005:**
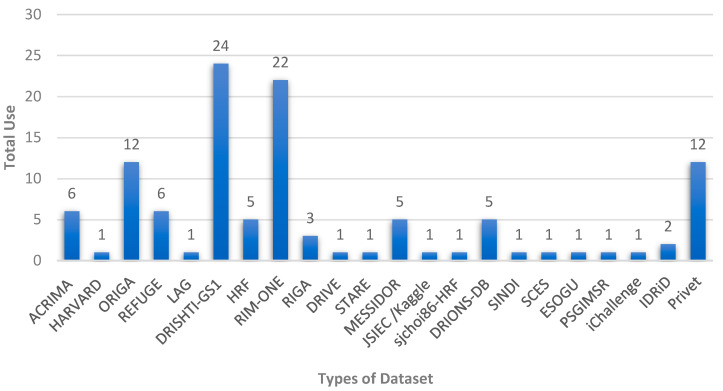
Different public datasets available for automated glaucoma diagnosis systems.

**Figure 6 diagnostics-13-02180-f006:**
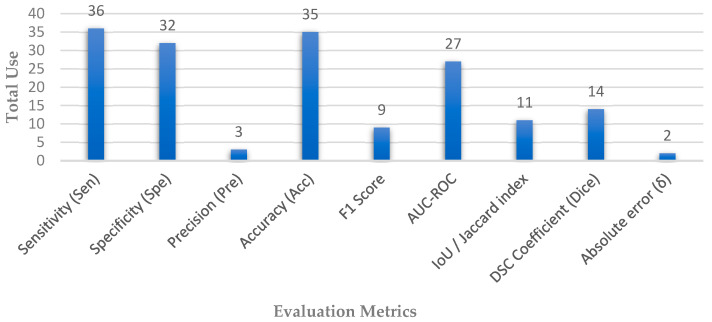
Distribution of evaluation metrics used in automated glaucoma diagnosis systems.

**Figure 7 diagnostics-13-02180-f007:**
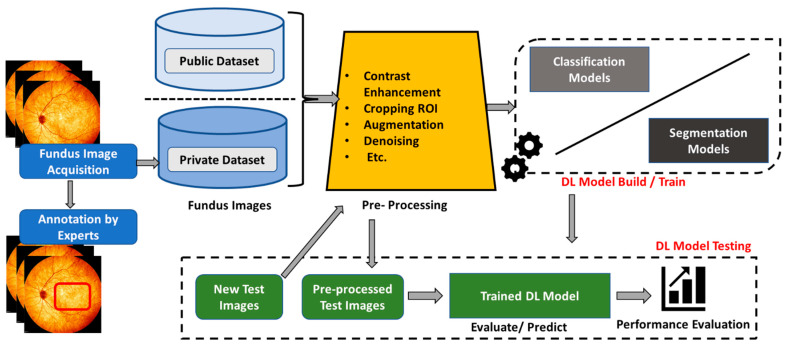
Overall deep learning framework for glaucoma diagnosis.

**Table 1 diagnostics-13-02180-t001:** Overview of fundus image datasets for glaucoma diagnosis.

Dataset	No. of Images	Glaucoma	Normal	Image Size	Cls. *	Seg.	Ground Truth Label
ACRIMA [22]	705	396	309	-	✔	-	-
HARVARD [23]	1542	756	786	-	✔	-	-
ORIGA [24]	650	168	482	3072 × 2048	✔	-	-
REFUGE [25]	1200	120	1080	2124 × 2056 1634 × 1634	✔	✔	Location of Fovea
LAG [26]	5824	2392	3432	3456 × 5184	✔	-	Attention maps
DRISHTI-GS1 [27]	101	70	31	2896 × 1944	✔	✔	CDR values and Disc center
HRF [28]	45	27	18	3504 × 2336	✔	✔	Center and Radius for Optic Disc
RIM-ONE-r1 [29]	169	51	118	-	✔	✔	-
RIM-ONE-r2	455	200	255	-	✔	-	-
RIM-ONE-r3	159	74	85	2144 × 1424	✔	✔	-
RIM-ONE-DL	485	172	313	-	✔	✔	-
RIGA [30]	750	-	-	2240 × 1488 2743 × 1936	-	-	Provide six boundaries for Optic Disc and Optic Cup
DRIVE [31]	40	-	-	768 × 584	-	✔	Vessel Segmentation
STARE [31]	402	-	-	605 × 700	-	✔	-
MESSIDOR [32]	1200	-	-	1440 × 960 2440 × 1488 2304 × 1536	-	-	Macular edema information
JSIEC/Kaggle [33]	51	13	38	-	✔	-	-
CHASE [3]	28	-	-	1280 × 960		✔	Vessel Segmentation
sjchoi86-HRF [34]	401	101	300	-	✔	-	-
DRIONS-DB [35]	110	-	-	600 × 400	-	-	Contours for the Optic Disc
SINDI [36]	5783	113	5670	-	✔	-	-
SCES [37]	1676	46	1630	3072 × 2048	✔	-	-
G1020 [38]	1020	296	724	-	✔	-	-
PAPILA [39]	488	155	333	2576 × 1934	-	✔	-
ESOGU [40]	4725	320	4405	-	✔	-	-
PSGIMSR [41]	1150	650	500	720 × 576	-	-	-

* Cls.: Classification, Seg.: Segmentation.

**Table 2 diagnostics-13-02180-t002:** General pre-processing methods for fundus images.

Pre-Processing Technique	Explanation
Contrast Enhancement (Histogram equalization)	Histogram equalization is a technique used to enhance the overall contrast of an image. The main objective of this method is to distribute the pixel values in an image’s histogram more evenly. The underlying principle of this technique is that if the distribution of pixel intensity values is more uniform, the resulting image will have increased contrast and will appear more visually appealing.
Contrast Enhancement (CLAHE)	CLAHE (Contrast Limited Adaptive Histogram Equalization) is a technique that improves image contrast by redistributing pixel intensity values. It adapts to local contrast, which will enhance the structures visibility in low-contrast images. CLAHE adjusts contrast of small image regions to avoid over-amplifying noise, which is useful for improving image quality in low-contrast settings.
Color Space Transformation	Color space transformation is a technique that improves image analysis accuracy by converting images from one color space to another. In fundus image analysis, the commonly used color spaces are RGB, HSI, and Lab.
Noise Removal	Noise removal is a crucial step in improving digital images’ quality by eliminating unwanted noise. Techniques such as median filtering, Gaussian filtering, and wavelet transform are commonly used to remove noise from images, thus improving image analysis accuracy.
Cropping of ROI	Cropping ROI is a technique used to isolate regions of interest in an image, such as the optic disc, macula, and blood vessels. This technique can improve the subsequent analysis accuracy by focusing on the relevant structures in the image.
Data Augmentation	Data augmentation is a technique used to increase the diversity of the training dataset by creating new variations of existing images. Some examples of this technique, which are used to enhance data variation, including rotation, flipping, scaling, and adding noise. This technique can improve deep learning models’ performance during the training phase by providing more diverse and representative dataset.

**Table 3 diagnostics-13-02180-t003:** Evaluation metrics for automated glaucoma screening and diagnosis systems.

Metric	Formula	Description
Sensitivity (Sen) Recall	Sen = TP */(TP + FN)	Measure of the percentage of patients with glaucoma who are correctly detected or identified by the model.
Specificity (Spe)	Spe = TN/(TN + FP)	Measures the proportion of patients without glaucoma who are correctly identified by the model.
Precision (Pre)	Pre = TP/(TP + FP)	Measures the proportion of patients identified as having glaucoma by the model who actually have the disease.
Accuracy (Acc)	Acc = (TP + TN)/(TP + TN + FP + FN)	Measures the proportion of patients who are correctly classified by the model.
F1 Score	F1 = 2 TP/(2 TP + FP + FN)	Measures the overall performance of the model in identifying both positive and negative instances.
AUC-ROC	The plot of the sensitivity against (1-specificity)	Measures the ability of the model to discriminate between patients with glaucoma and those without the disease.
IoU/Jaccard index	IoU = TP/(TP + FN + FP)	Measures the overlap between the predicted and ground truth segmentation masks.
DSC Coefficient (Dice)	Dice = 2 TP/(2 TP + FP + FN)	Measures the similarity between the predicted and ground truth segmentation masks.
Absolute error (δ)	δ = |CDR_p_ − CDR_g_|	Measures the absolute error where CDRp and CDRg denote the cup to disc ratio value for the prediction and ground truth, respectively.

* TP: True Positive, FN: False Negative, TN: True Negative, FP: False Positive.

**Table 5 diagnostics-13-02180-t005:** Summary on glaucoma diagnosis systems.

Reference	Dataset Description	Architecture	Strengths	Limitations
Li et al. [26]	A private dataset (LAG), 11,760 images	ResNet	The model consists of three subnets that detect glaucoma using deep features from visualized maps of pathological areas.	The model is complex and requires complex mathematical operations.
Wang et al. [56]	DRIONS-DB, HRF, RIM-ONE, and DRISHTI-GS 1), 686 images	VGG & AlexNet	The 3D topographic map of the ONH, reconstructed using the shape from shading method, offers improved visualization of the OC and OD.	The model is complex and resource- intensive.
Gheisari et al. [57]	A private dataset, 695 images	VGG & ResNet & RNN (LSTM)	Extraction of spatial and temporal data from fundus videos is much more accurate when CNN and RNN are used in a single system.	Despite its high accuracy, further evaluation of a larger heterogeneous population is required.
Nayak et al. [58]	A private dataset, 1426 images	CNN	A feature extraction technique that uses a meta-heuristic approach, requiring fewer parameters for efficient feature learning.	The developed model is incapable of automatically detecting different stages of glaucoma.
Li et al. [59]	A private dataset, 26,585 images	ResNet	Integrating fundus images with medical history data slightly improves sensitivity and specificity.	Bias was introduced by subjective grading from two groups, and cropping the optic nerve head region may cause information loss.
Hemelings et al. [60]	A private dataset, 8433 images	ResNet	By combining transfer learning, careful data augmentation, and uncertainty sampling, labelling costs were reduced by about 60%.	Considerations include imbalanced data, late-stage glaucoma images, and primarily Caucasian patients in the models.
Juneja et al. [61]	DRISHTI-GS and RIM-ONE, 267 images	CNN	By using separable convolutional layers, increasing filter size led to a more accurate classification.	The model’s use of manual cropping of the optic disc leads to data loss.
Liu et al. [62]	A private dataset, 241,032 images	ResNet	An online DL system was proposed that updates the model iteratively and automatically using a large-scale database of fundus images.	The model’s generalization ability can be enhanced by human–computer interaction.
Bajwa et al. [63]	ORIGA, HRF, and OCT & CFI, 780 images	VGG16 & CNN	An automated disc localization model was created using a semi-automatic method for generating ground truth annotations, facilitating classification.	The proposed network struggles with learning distinctive features to classify glaucomatous images in public datasets.
Kim et al. [64]	A private dataset, 2123 images	VGGNet, InceptionNet & ResNet	A weakly supervised localization method highlights glaucomatous areas in input images. A prototype web app for diagnosis and localization of glaucoma was presented, integrating the predictive model and publicly available.	Using an external dataset produced lower accuracy scores compared to using the dedicated dataset during training in the experiments.
Hung et al. [65]	A private dataset, 1851 images	EfficientNet	Evaluation methods differed based on binary and ternary classifications, the use of red-free and non-red-free photographs, and the inclusion of high myopia information.	Limitations include a small number of cases, a single ethnic background, and the exclusion of pre-perimetric glaucoma.
Cho et al. [66]	A private dataset, 3460 images	InceptionNet	Averaging multiple CNN models with diverse learning conditions and characteristics is more effective in classifying glaucoma stages compared to using a single CNN model.	More diverse data are needed for a generalized model, and further studies are necessary to adjust weighted values per model and improve performance.
Leonardo et al. [67]	ORIGA, DRISHTI-GS, REFUGE, RIM-ONE (r1, r2, r3), and ACRIMA, 3187 images	EfficientNet, U-Net	Using GAN to improve quantitative and qualitative image quality, and proposing a new model to evaluate the quality of fundus images.	Generative model and quality evaluator were trained with full field-of-view images, while lower field-of-view images are common in different equipment and datasets.
Alghamdi et al. [68]	RIM-ONE and RIGA, 1205 images	VGG-16	Comparing the performances of three automated glaucoma classification systems (supervised, transfer, and semi-supervised) on multiple public datasets.	The proposed models can only diagnose the presence or absence of glaucoma and cannot classify the severity of a specific retinal disease.
Devecioglu et al. [40]	ACRIMA, RIM-ONE, and ESOGU, 5885 images	Self-ONNs	Self-ONNs show high performance in glaucoma detection with reduced complexity compared to deep CNN models, especially with limited data.	The suggested model does not include a segmentation network.
De Sales et al. [70]	DRISHTI-GS and RIM-ONEv2, 556 images	VGG16	The use of 3DCNN resulted in high accuracy and the production of 3D activation maps, which provide additional data details without the need for optic disc segmentation or data augmentation.	It requires more parameters compared to 2D convolution, making it computationally more expensive and technically challenging.
Joshi et al. [41]	DRISHTI-GS, HRF, and DRIONS-DB and one privet dataset PSGIMSR, 1391 images	VGG &ResNet & GooglNet	Ensembling pre-trained individual models using a voting system can improve the accuracy of the proposed diagnosis model.	The proposed framework does not include the segmentation of the OD and OC, which could potentially increase the detection performance.
Almansour et al. [71]	RIGA, HRF, Kaggle, ORIGA, and Eyepacs and one privet dataset, 3771 images	R-CNN & VGG	Proposing a two-step approach for early diagnosis of glaucoma based on PPA in fundus images using two localization and classification models.	It uses a complex deep learning system to classify PPA versus non-PPA; therefore, an interpretable surrogate model can be used.
Liao et al. [74]	ORIGA, 650 images	ResNet	The proposed framework addresses the issue of interpretability in deep learning-based glaucoma diagnosis systems by highlighting the specific regions identified by the network.	Although the method shows good accuracy, it still suffers from the problem of high-resolution feature maps being hard to represent by the proposed method.
Sudhan et al. [75]	ORIGA, 650 images	U-Net & DenseNet-201 and DCNN	This model can be useful for various medical image segmentation and classification processes such as diabetic retinopathy, brain tumor detection, breast cancer detection, etc.	The performance of the proposed system can be enhanced by solving the imbalance issue by improving the classifier and reducing the threshold.
Nawaz et al. [76]	ORIGA, 650 images	EfficientNet-B0	Proposing a robust model based on the EfficientNet-B0 for key points extraction to enhance the glaucoma recognition performance while decreasing the model training and execution time.	More robust feature selection methods can be implemented and employed in deep learning models to expand this work to other eye diseases.
Diaz-Pinto et al. [77]	ACRIMA, DRISHTI GS1, sjcho 86-HRF, RIM-ONE, HRF, 1707 images	VGG16, VGG19, InceptionV3, ResNet50, and Xception	The study evaluated five ImageNet-trained CNN architectures as classifiers for glaucoma detection and found them to be a reliable option with high accuracy, specificity, and sensitivity.	CNN models’ performance can decrease when tested on databases not used in training, and varying labeling criteria across publicly available databases can impact classification results.
Serte et al. [78]	HARVARD, 1542 images	AlexNet, ResNet-50, and ResNet-152	A graph theory-based technique is recommended for identifying salient regions in fundus images by locating theptic disc and removing extraneous areas, accompanied by an ensemble CNN model to enhance classification accuracy.	A limitation of the approach is the absence of a segmentation process prior to classification, which could potentially result in less accurate outcomes.
Jos´e et al. [79]	ORIGA, DRISHTI-GS, RIM-ONE-r1, RIM-ONE-r2, RIM-ONE-r3, iChallenge, and RIGA, 3231 images	Multi-scale encoder— decoder network and Mobile Net	The newly created pipeline can enable large-scale glaucoma screenings in environments where it was previously impractical because of its capability to operate without an Internet connection and run on low-cost mobile devices.	The dataset utilized in this study poses some challenges, including imbalanced classes and a limited number of samples for deep learning techniques.
Natarajan et al. [7]	ACRIMA, Drishti- GS 1, RIM-ONEv1, and RIM-ONEv2, 2180 images	SqueezeNet	A highly accurate, lightweight glaucoma detection model has been introduced. The model’s stages can be used separately or with other models through transfer learning for future ocular disorder diagnosis and treatment frameworks.	The model only uses deep features and does not consider geometric or chromatic measures in the disc and cup region, but the classifier output does not provide insights for ophthalmologists.
Islam et al. [73]	HRF and ACRIMA and one privet dataset BEH, 1188 images	U-Net, EfficientNet, MobileNet, DenseNet, and GoogLeNe	A new dataset for identifying glaucoma with lower training time was developed by segmenting blood vessels from retinal fundus images using the U-Net model.	The accuracy of the blood vessel segmentation model is slightly lower compared to the segmentation of the optic cup and optic disc.

**Table 7 diagnostics-13-02180-t007:** Summary on OD/OC segmentation systems.

Reference	Dataset Description	Architecture	Strengths	Limitations
Civit-Masot et al. [80]	DRISHTI-GS, and RIM-ONE v3, 136 images	U-Net, Mobile Net	The implementation used a lightweight MobileNet for embedded model deployment, and a reporting tool was created to aid physicians in decision-making.	There is a need to train models using larger datasets from both public and private sources.
Pascal et al. [81]	REFUGE, 1200 fundus images	U-Net (VGG)	The study employs a single DL architecture and multi-task learning to perform glaucoma detection, fovea location, and OD/OC segmentation with limited resources and small sample sizes.	The use of shared models requires additional effort from experts, and obtaining pixel-wise annotations of objects for segmentation and fovea location is more time consuming and expensive.
Fu et al. [83]	ORIGA, SCES, and SINDI, 8109 images	U-Net ResNet	A segmentation-guided network is employed to localize the disc region and generate screening results, while a pixel-wise polar transformation enhances deep feature representation by converting the image to the polar coordinate system.	Despite using a complex ensemble system consisting of multiple layers and transformations for glaucoma detection, the results are not as high as simpler algorithms that produce better outcomes.
Sreng et al. [84]	REFUG, ACRIMA, ORIGA, RIM–ONE, and DRISHTI–GS1, 2787 images	DeepLabv3+ (Mobile Net) AlexNet, Google Net, InceptionV3, XceptionNet, Resnet, ShufieNet, SqueezeNet, MobileNet, InceptionResNet, & DenseNet	The proposed framework employs five deep CNNs for OD segmentation, eleven pretrained CNNs using transfer learning for glaucoma classification, and an SVM classifier for optimal decision-making.	The study used high-quality images, emphasizing the need for a representative dataset with co-morbidities and low-quality images. In addition, the proposed segmentation method is limited to OD only, and further development is required to segment both OD and OC for improved classification accuracy.
Yu et al. [85]	ORIGA, RIM–ONE, and DRISHTI–GS1, 882 images	U-Net (ResNet) ReNeXt (ResNet)	The proposed segmentation/classification model uses a pre-trained network for fast training and a morphological post-processing module to refine the optic disc and cup segmentations based on the largest segmented blobs.	Model performance is impacted by low-quality images and severe disc atrophy, requiring training on images with less perfect quality and pathological discs with atrophy to improve segmentation.
Natarajan et al. [86]	DRIONS, 2311 images	MKFCM VGG	This research employs Modified Kernel Fuzzy C-Means (MKFCM) clustering for optimal clustering of retinal images, achieving accuracy even in noisy or corrupted input images.	Data augmentation can improve algorithm performance, making it applicable to various glaucoma retinal diseases, including cases where healthy images are misclassified as mild or glaucomatous.
Ganesh et al. [87]	ACRIMA, DRISHTI-GS, RIM-ONEv1, REFUGE, and RIM-ONEv2, 1830 images	U-Net (inception)	The proposed system integrates segmentation and classification within a single framework and utilizes inception blocks in the encoder and decoder blocks of the GD-YNet to capture features at multiple scales.	The ResNeXt block architecture is restricted by assuming a cardinality value of 32, and the encoder–decoder paths have basic inception modules with dimension-reduction capabilities, which is another limitation.
Juneja et al. [88]	DRISHTI-GS, 101 images	U-Net	The proposed architecture utilizes two neural networks in tandem, with the second network building upon the processed output of the first. This concatenated network achieves superior accuracy with reduced complexity.	The proposed system only outputs the CDR and does not provide information about the severity level of the disease.
Tabassum et al. [90]	DRISHTI-GS, RIM-ONE, and REFUGE, 660 images	U-Net (CDED-Net)	The proposed encoder–decoder design is computationally efficient and eliminates the need for pre/post-processing steps by reusing information in the decoder stage and using a shallower network structure.	The algorithm has limitations that require further research and testing with diverse data. Additionally, its effectiveness in diagnosing other retinal diseases needs to be verified.
Liu et al. [91]	DRISHTI-GS and REFUGE, 1301 images	U-Net (DDSC-Net)	The network uses a deep separable convolution network with dense connections and a multi-scale image pyramid at the input end.	The model may not provide consistent results for fundus images captured by different devices and institutions, indicating a generalization issue.
Nazir et al. [92]	ORIGA, 650 images	Mask-RCNN	The proposed method accurately segments OD and OC regions in retinal images for glaucoma diagnosis, even with image blurring, noise, and lighting variations.	The proposed model, based on R-CNN, could be improved by incorporating newer deep learning techniques such as EfficientNet.
Rakhshanda et al. [93]	RIM–ONE v3, and DRISHTI–GS1, 260 images	encoder–decoder Network & VGG16	The study combines semantic segmentation and medical image segmentation to enhance performance and reduce memory usage and training/inference times.	Semantic segmentation has a limitation in that it cannot distinguish between adjacent objects, such as OD and OC.
Wang et al. [94]	MESSIDOR, ORIGA, and REFUGE, 1970 images	Multi-scale encoder— decoder network, (Modified U-Net)	The proposed network uses multiple multi-scale input features to counteract pooling operations and preserve important image data. Integration via element-wise subtraction highlights shape and boundary changes for precise object segmentation.	Limitations include the model’s sensitivity to object boundaries and small image sizes used in experiments due to GPU memory constraints.
Kumar et al. [95]	BinRushed, Magrabia, MESSIDOR, DRIONS-DB, RIM-ONE, and IDRiD, 1839 images	U-Net	The study utilized a modified U-Net architecture with 19 layers and implemented ground truth generation to improve the model’s training and testing procedures.	Transfer Learning or GANs can potentially reduce the time required to build the model.
Panda et al. [96]	ORIGA, RIM–ONE, and DRISHTI–GS1, 882 images	Residual DL	The proposed technique has the potential to assist doctors in making highly accurate decisions regarding glaucoma assessment in mass screening programs conducted in suburban or peripheral clinical settings.	Incorporating post-processing techniques to obtain the segmentation output adds an extra step that increases the model’s complexity.
Fu et al. [97]	DRIVE, Kaggle, MESSIDOR, and NIVE, 11,640 images	U-Net	The proposed method combines a data-driven U-Net and model-driven probability bubbles to locate the OD, resulting in a more robust joint probability approach for localization, ensuring effectiveness.	This paper’s scope is limited to interactions between the model-driven probability bubble approach and the deep network’s output layer, without covering deeper interactions in hidden layers.
Zhao et al. [98]	DRIONS-DB and DRISHTI-GS, 211 images	U-Net with transfer learning	The proposed algorithm can effectively segment OD/OC in fundus images, even with only a small number of labels. While providing fast segmentation, the method also maintains a relatively high level of accuracy.	Compared to existing algorithms used for comparison, the proposed method demonstrates an OD/OC segmentation accuracy that is only slightly lower by less than 3%.
Hu et al. [99]	REFUG, MESSIDOR, RIM-ONE-r3, DRISHTI-GS, and IDRiD, 2741 images	Encoder- decoder	The proposed model effectively handles issues caused by domain shifts from different acquisition devices and limited sample datasets, which may result in inadequate training.	The proposed model did not address the issue of the blurred boundary between the OD and OC.
Baixin et al. [100]	MESSIDOR and RIM-ONE-r1, 1369 images	U-Net	The proposed method enhances semantic segmentation with channel dependencies and integrates multi-scale data into the attention mechanism to utilize contextual information.	The suggested method did not perform as well as previous methods, possibly due to factors such as network design, data pre-processing, and hyperparameter adjustment.
Shankaranarayana et al. [101]	ORIGA, RIMONE r3, and DRISHTI–GS1, 910 images	Encoder decoder	The proposed pretraining scheme outperforms the standard denoising autoencoder. It is also adaptable to different semantic segmentation tasks.	Although increasing the batch size during training may enhance the performance of the proposed models, it was not feasible in the study due to system limitations.
Bengani et al. [102]	DRISHTI GS1, RIM-ONE, and Kaggle’s DR, 88,962 images	Encoder- Decoder	This model trains quickly and has a small disk space requirement compared to other models that exhibit similar performance.	The study did not experiment with any method to train the autoencoder and segmentation network simultaneously.
Wang et al. [103]	DRISHTI-GS dataset, RIM-ONE-r3, and the REFUGE, 1460 images	DeepLabv3+ and MobileNetV2	It proposed a new segmentation network with a morphology-aware loss function to produce accurate optic disc and optic cup segmentation results by guiding the network to capture smoothness priors in the masks.	No domain generalization techniques were used in this study to address the problem of retraining a new network when the image comes from a new target domain.

**Table 8 diagnostics-13-02180-t008:** Research gaps.

Process	Gap
Datasets	Private datasets pose limitations in research, as they hinder accurate assessment by making it challenging to compare the results of different datasets. Moreover, some methods may utilize unsuitable datasets, leading to authors generating private ground truths.
Enhancement	Image enhancement algorithms can cause artifacts and distortions, making the image unusable. In addition, parameter selection is subjective and can vary depending on the application and preference; however, these techniques can also be computationally expensive.
Localization	Using only the bright circular region principle for localizing the optic disc in retinal fundus images is inaccurate due to other bright areas in the image. Some methods also require manual intervention, such as manual annotation of the disc area and its radius.
Segmentation	Typically, when using the same segmentation method for both the optic disc and optic cup, the results for the latter are not as satisfactory. This is due to disregarding the relationship between different parts of the retina in most of the proposed methods.
Classification	The accuracy of separating different parts of the retina depends heavily on the extracted features, which are crucially dependent on several parameters. However, most methods only consider a few of these parameters, and large datasets are also required for optimal network fitting. Unfortunately, such datasets are not readily available for glaucoma patients, as the training procedures are time-consuming.

## Data Availability

Not applicable.

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
