# Peer review of "Automated Glaucoma Screening and Diagnosis Based on Retinal Fundus Images Using Deep Learning Approaches: A Comprehensive Review"

_diagnostics, 2023, doi:10.3390/diagnostics13132180_

Round 1
Reviewer 1 Report
In this review, the authors aimed to to present a comprehensive overview of the most recent techniques for detecting glaucoma by means of deep learning methods applied to retinal fundus images.
Overall I found the paper to be well written and the methodology well described. Clinical data is clearly presented. There are many values in the work presented, but the manuscript suffers from some minor issues, which need to be addressed.
1- In the inclusion criteria section please indicate these points:
a) Were all the included studies treatment naïve
b) Were all the included studies on any type of glaucoma? or they considered some specific types of glaucoma, like primery or open angle or close angle
2- One of the important limitation was to use different fundus photo machine in different studies which should be mentioned
Reviewer 2 Report
A Comprehensive analysis of various deep learning methods for glaucoma diagnosis is presented in this paper. The paper organization and written is well enough as a review paper.
1-Fig 3 :Relationship between the collected articles with respect to the years of publication, The caption is in arabic,?!!!
2-In Table 1, the image sizes for dataset are presented but in glaucoma diagnosis systems (such that shown in Table 5), no sign of this important factor is found. This factor should be taken into account and may change the evaluation metrics that presented in Table 3.
3- Since, the paper concentrates on deep learning, and based on the review, in discussion and conclusion, their is no recommendation about which of the deep neural networks (VGG & ResNet, & GooglNet, etc) perform best for diagnosis !!
4-In conclusion the author states:" integrating DL model into clinical practice remains a big future challenge" , but he never stated any thing about this challenge, and how to deal with? and how other deals with?
Reviewer 3 Report
Reviewer’s comments
Automated Glaucoma Screening and Diagnosis Based on Retinal Fundus Images using Deep Learning Approaches: A Comprehensive Review
Glaucoma is the second leading cause of blindness worldwide after Cataract which shares almost 30 percent of the total quantum of preventable blindness. Glaucoma's prevalence ranges from 2 to 5 percent worldwide with a predominance of Primary open-angle glaucoma cases except in Mongolia, Myanmar, and northern India where primary angle-closure glaucoma (PACG) has been reportedly more common. Of 80 million known glaucoma patients globally, the Asian ethnicity contributes almost half of this volume including 16 million from India. Malays are the third largest ethnic group in Asia. The prevalence of known glaucoma patients among them is estimated at around 2.4 percent. The prevalence of glaucoma among Chinese is approximately 3.2 percent in the age group of 40 to 80 years whereas Glaucoma shares 9 to 12 percent cases of preventable blindness in the USA.
The prevalence of known Glaucoma in African races is around 6%, which is much higher compared compare to other races. Over and scanty infrastructure and adequate access to the facilities make it impossible to assessthe exact prevalence of glaucoma in Africa. It is worth considering that an additional 120 million people are expected to have ocular hypertension, primary angle-closure suspects (PACSs), or PAC. The global engagement of Glaucoma is likely to be more than 112 million people within the next 20 years. Hence it is highly prudent to develop a mechanism for early detection as well as unmask unknown glaucoma patients by sustained efforts to galvanize the process of screening in vulnerable populations worldwide. The process of glaucoma detection as well as subsequent monitoring of arrest of progression of glaucoma amongthe population needs the expertise of highly skilled ophthalmologists and equipment like a visual field analyzer, OCT, and fundus camera. It is practically impossible to extend such facilities in every corner of the globe in the near future due to the obvious lack of resources. Hence the role of effective application of various modes of artificial intelligence including the Deep learning process is of immense significance to a content load of preventable blindness , especially in cases of glaucoma.
Deep learning has gained wide acceptance in scientific computing application of its algorithms in various fields including industries, health care, metrology, the agriculture sector, space, security, and intelligence to solve complex problems. All deep learning algorithms use different types of neural networks to perform specific tasks. An artificial neural network also known as nodes is a complex hardware specifically integrated to operate like a human brain. These nodes are configured in three layers, The input layer, the hidden layer, and the output layer. Neural networks can identify and Translate text, Identify faces, Recognize speech, Read handwritten text, identify colours,photographs , temperature, humidity control robots .There are numerous and multiple types of Algorithms in use in the process of Deep Learning. Among the most prominent are Convolutional Neural Networks (CNNs), Long Short Term Memory Networks (LSTMs), Recurrent Neural Networks (RNNs), Generative Adversarial Networks (GANs), Radial Basis Function Networks (RBFNs), Multilayer Perceptrons (MLPs), Self-Organizing Maps (SOMs), Deep Belief Networks (DBNs), Restricted Boltzmann Machines(RBMs)and Autoencoders.
Deep learning in healthcare offers path-breaking applications. Deep learning has been used in medical imaging in almost each and every specialty of medicine including MRI scans, CT scans, ECG, OCT, Visual field analysis, and many more. Deep learning technique is also used in genomics to understand a genome. Deep learning gathers a massive volume of data, including patients’ records, medical reports, investigations, and imaging records, and applies its neural networks to provide highly useful and precise interpretation, building end-to-end systems to analyze disease patterns so as to facilitate better medical management.
Reviewer is in consonance with authors that linking deep learning process with automated glaucoma diagnosis will definitely enhance the accuracy and precision to facilitate early detection and monitoring of Glaucoma among larger section of the population.
reviewer compliments the authors for this very useful study.
